# Frailty Assessment and Its Impact on Loneliness among Older Adults Receiving Home-Based Healthcare during the COVID-19 Pandemic

**DOI:** 10.3390/healthcare12161666

**Published:** 2024-08-21

**Authors:** Maria Klesiora, Konstantinos Tsaras, Ioanna V. Papathanasiou, Maria Malliarou, Nikolaos Bakalis, Lambrini Kourkouta, Christos Melas, Christos Kleisiaris

**Affiliations:** 1Department of Nursing, University of Thessaly, Gaiopolis, 41500 Larissa, Greece; mklesiora@uth.gr (M.K.); ktsa@uth.gr (K.T.); iopapathanasiou@uth.gr (I.V.P.); malliarou@uth.gr (M.M.); 2Department of Nursing, University of Patras, 26334 Patras, Greece; nikosbakalis@upatras.gr; 3Department of Nursing, International Hellenic University, Sindos, 57400 Thessaloniki, Greece; lkourkouta@nurse.teithe.gr; 4Department of Nursing, Hellenic Mediterranean University, 71410 Heraklion, Greece

**Keywords:** frailty, loneliness, social isolation, older adults, COVID-19 pandemic, home care, ageing

## Abstract

The purpose of this study was to identify the effect of frailty on loneliness among older adults receiving home care, in correlation to their socioeconomic and homebound statuses. This cross-sectional study recruited 218 individuals aged ≥65 years receiving home-based health services from the “Help at Home” program in the Region of Evrytania, Greece through an open invitation from the municipality authorities from March to June 2022. The Tilburg Frailty Indicator (TFI) was used for the evaluation of frailty, the UCLA Loneliness Scale version 3 was used for loneliness, and social isolation was accessed through five questions (living alone, frequency of interaction with children, relatives, friends, and participation in social organizations). The mean age of the participants was 81.48 ± 9.06, 61.9% were female, 54.1% experienced high levels of loneliness (UCLA-3 mean 45.76 ± 11.10 [range 20–68]), and 46.3% of the participants were found to be socially isolated. Also, 58.3% of the individuals were identified as frail (TFI mean 5.95 ± 3.07) [TFI range 0–13], with 57.3% experiencing physical frailty, 43.6% experiencing psychological frailty, and 27.1% experiencing social frailty. An analysis of covariance (ANCOVA) using UCLA-3 as the dependent variable revealed that loneliness across all domains of TFI was significantly higher in participants with frailty (total frailty [Yes] 49.27 vs. [No] 40,87 *p* < 0.001) (physical frailty [Yes] = 48.99 vs. [No] = 41.42, *p* < 0.001, psychological: 48.60 vs. 43.57 *p* < 0.001, and social: 53.38 vs. 42.94 *p* < 0.001), particularly compared to non-frail individuals, even after adjusting for potential confounding effects (covariates: gender, age, marital status, family status, living status, educational level, annual income, chronic diseases, homebound status, and social isolation). Our findings indicate that frail older adults experienced higher levels of loneliness, suggesting that frailty and loneliness are independently associated among older adults who receive home-based healthcare.

## 1. Introduction

Loneliness in older adults has emerged as an urgent public health concern, owing to its recent high prevalence and the correlated adverse health impacts in old age. Interestingly, the prevalence of loneliness in people ≥60 years old in European countries is 11.9–19.3% [1]. Especially in old age, various factors commonly linked to the person’s social disadvantages (living alone, low educational level, and restricted social connections) and psychological elements (depressive symptoms), as well as poor quality of life, have been identified as contributing to an elevated sense of loneliness [2]. In addition, the risk of mortality due to loneliness is comparable to well-documented clinical risk factors, such as smoking and high blood pressure [3]. It has also been suggested that loneliness is strongly related to substantial health repercussions, encompassing physiological effects such as obesity, cardiovascular issues, sleep disturbances, and weakened immunity, as well as psychological effects, including depression, social anxiety, and cognitive decline [4].

Most importantly, loneliness represents a subjective feeling of discontentment stemming from the individual’s perception that social relationships are unsatisfactory in either quantity or quality [5]. Thus, it is important to distinguish loneliness from social isolation, which refers to one’s objectively limited network of relationships. These two conditions may co-exist independently. For instance, someone might feel lonely despite having many social connections or might not experience loneliness even if they have limited social interconnections [6,7].

On the other hand, frailty is also frequent among community-dwelling people aged 75 years or older and leads to poorer quality of life [8], especially for those receiving home care, with a frequency of 14% to 38% [9].

Frailty is widely recognized as a gradual decline in the functioning of physiological systems associated with increased age, leading to negative health outcomes [10], particularly restrictions in mobility, disability, increased risk of cognitive declines, falls, hospitalization, and premature deaths [11,12,13]. Considering that several European projects under the European platform of WE4AHA & EIPonAHA have extensively investigated the multidimensional nature of frailty in recent years, the assessment of physical, psychological, and social domains, including sociodemographic factors (age, marital status, education, and income) that increase the risk for frailty development, has become urgent [14].

There has been growing concern globally about the association between frailty and loneliness as shared comparable psycho-socio-economic factors among community-dwelling older adults [15], highlighting the significance of examining frailty by employing a multidimensional approach and the importance of social determinants on physical health in aging [16,17], demonstrating that a bidirectional relationship may exist between loneliness and frailty. It has been demonstrated that older people receiving residential care or receiving home care services experienced the highest levels of anxiety and loneliness during the first wave of the COVID-19 pandemic [18]. In addition, frailty has been found to contribute to physical loneliness among older adults receiving home care, who, as a consequence, may not be able to make substantial progress towards well-being and health [19]. Therefore, the importance of creating new concepts and structures regarding frailness is highlighted. These will not only assist in addressing loneliness among older, home-dwelling people by addressing the issues with home care nursing and meeting their need for social contact [20] but also, in distinguishing the different domains in which frailty provides valuable information about older adults’ needs, will allow care providers to anticipate the increasing number of independently living older adults and deliver to them tailored care and support, which can contribute to their independent living situation and well-being [21].

In Greece, the prevalence, risk factors, and the relationship between frailty and socioeconomic factors and the quality of life in a community-dwelling population have been extensively investigated [22,23,24,25]. However, to our best knowledge, there are no studies available that have examined the epidemiological association between frailty and loneliness in the context of socioeconomic status in older adults receiving home-based healthcare services. From this perspective, the widespread occurrence of frailty in older adults, along with its various impacts on daily life and multidimensional nature, makes it important to explore how frailty may contribute to increased loneliness. Understanding this link could highlight the potential for preventive interventions targeting frailty to alleviate loneliness among seniors. Thus, the exploration of socioeconomic factors in association with higher levels of loneliness will help in identifying and implementing appropriate and effective interventions in home care settings.

Therefore, in this survey, the investigation of the existing gap regarding the lack of home-based studies was mainly designed to draw attention to the problem of loneliness in older adults, in correlation to frailty, existing comorbidities, and socioeconomic factors.

## 2. Materials and Methods

### 2.1. Study Design

A cross-sectional study was conducted from March to June 2022 in the region of Evrytania, Greece. The study recruited individuals aged 65 years old and over who were registered members of the home-based community program “Help at Home”. This home care program provides domestic assistance to elderly people who live alone constantly or for some hours a day and cannot adequately take care of themselves, as well as to disabled people who face situations of isolation. Criteria to register and benefit from the program are: old-age, invalidity or survivor’s pension provided by insurance organizations incorporated in the e-National Social Security Fund, and retired civil servants on old-age and invalidity or survivor’s pension [26]. In general, the recipients of “Help at Home” services are mainly older adults with disadvantaged social status of poor family support who receive medical, nursing, and social care. In this convenience sampling study, data were collected by the main researcher (psychologist) through door-to-door screenings focused on frailty, loneliness, and social isolation according to the STROBE guidelines [27]. The sample size of the study was calculated by power analysis using G* Power 3 software (v3.1.9.7). Considering a medium effect size (f2 = 0.25), a significance level of 5% (alpha level), a statistical power of 95%, the number of comparison groups as 2, and the number of covariates as 10, a minimum sample size of 210 individuals was required. After an open invitation by the local municipality authorities, a total of 358 older adults (registered members of the “Help at Home” program) were screened. Members were excluded if they: (1) were aged ≤65 years old; (2) had severe visual or hearing impairments; (3) refused to participate in the questionnaire survey; or (4) could not communicate due to the severity of the chronic medical conditions. For instance, bedridden patients due to advanced stage of cancer, severity of stroke, and/or severe dementia diagnosis were excluded to avoid methodological issues linked to the “social isolation” or “decline in cognition” of the participants, based on their medical histories (medical records). Finally, 218 out of 358 registered individuals with full data met our eligibility criteria and were involved in the statistical analysis (response rate 60.9%).

### 2.2. Instruments

#### 2.2.1. Loneliness Assessment

The feeling of loneliness was investigated with the University of California, Los Angeles Loneliness Scale Version 3 (UCLA Loneliness Scale Version 3) [28]. This is a 20-item scale, and each item has four options to reflect the frequency (1 = never, 2 = rarely, 3 = sometimes, and 4 = often). The total score is calculated by adding the individual scores of each item, with 9 out of the 20 questions being reverse-scored. The total score ranges from 20 to 80, with the higher scores corresponding to a greater level of loneliness. In this study, to evaluate loneliness severity, the cut-off criteria were used as follows: <28 = no/low loneliness, 28–43 = moderate loneliness, and >43 = high loneliness [29]. The UCLA-3 scale was validated in Greek adults, showing strong correlations with the Hospital Anxiety Depression Scale (construct validity *p* < 0.05) and significant test–retest reliability for social loneliness at 0.826, for psychological loneliness at 0.980, and for isolation at 0.880 [30]. In our study, the UCLA-3 reliability was α = 0.952.

#### 2.2.2. Frailty Assessment

Given that the majority of the frailty instruments include only items on physical frailty characteristics and the multidimensional nature of both frailty and loneliness as shared common and comparable psycho-socio-economic factors, in this study, we evaluated frailty utilizing section B of the Tilburg Frailty Indicator (TFI), following the suggestions of Verver and his colleagues [21]. In particular, TFI comprises 15 elements organized into three distinct domains of frailty. The components of the physical domain are poor physical health, unintended weight loss, difficulty walking, maintaining balance, poor hearing, poor vision, lack of strength in the hands, and physical tiredness. The evaluation of the psychological domain consists of problems with memory, feeling down, feeling nervous or anxious, and inability to cope with problems. Finally, the social domain includes living alone, missing other people, and lack of social support. The scores range from 0 to 15 (total frailty), 0 to 8 (physical frailty), 0 to 4 (psychological frailty), and 0 to 3 (social frailty), with the higher scores referring to a higher level of frailty [31]. Total frailty is recognized if the total score is at least 5. In terms of individual domains, the thresholds are at least 3 out of 8 components for physical frailty, at least 2 out of 4 for psychological frailty, and at least 2 out of 3 for social frailty [32]. Part A of the TFI includes 10 determinants of frailty, but in this study, we collected the data of 6 determinants (age, sex, marital status, economic status, education, and presence of chronic diseases), as they are considered well-known sociodemographic determinants of both frailty and loneliness [2,33].

The TFI was recently validated for the Greek community-dwelling older people (mean age = 79.7 years), and its physical, psychological, and social domains were verified using Pearson correlation coefficients between the domains and alternative measures. Among other criteria, feeling lonely was also applied (De Jong Gierveld loneliness scale) as a criterion for its concurrent validity (≥0.70) [34]. In our study, the reliability of TFI was α = 0.744.

#### 2.2.3. Social Isolation

For the assessment of social isolation, we used an approach that was previously applied in recent British longitudinal studies [35,36]. One point was assigned for each of the following criteria: residing alone (unmarried or not cohabiting), having less than monthly contact with each child, other family members and friends (whether in person, through written communication, or via telephone), and not being affiliated with any kind of organizations or groups. These scores were totaled and ranged from 0 to 5. Higher totals indicated a greater degree of social isolation. For the regression analyses, we divided the participants into three categories according to whether their score was low (0), average (1), or high (≥2).

#### 2.2.4. Socioeconomic Assessments

Socioeconomic status was estimated by recording the individual characteristics of participants (sex, education level, marital status, having children, living alone, and annual individual income). In particular, an annual individual income of less than EUR 5.269 was considered the “poverty threshold”, according to the Hellenic Statistical Authority [37].

Homebound status was determined based on the ability of older adults to leave the home due to each illness in the last month as follows: (1) homebound—individuals never or rarely left the home during the last month; (2) semi-homebound—individuals had difficulty leaving the home (a few times per week) but only with assistance; and (3) non-homebound—individuals who had no difficulty or need for help to leave the house at least twice per week [38].

Comorbidity was assessed by recording the most common chronic diseases of the participants according to their medical histories, such as cancer diagnosis, history of cardiovascular diseases, stroke, chronic obstructive pulmonary disease, hypertension, diabetes mellitus, arthritis, kidney disease, etc. Multimorbidity was defined as a documented history of at least two chronic diseases and categorized as follows: 0, 1, 2, and at least 3 [39].

### 2.3. Ethical Consideration

This study was ethically approved by the Scientific Committee of the MSc Program “Management of Aging and Chronic Diseases” of the Hellenic Open University (IRB: Pr No 7982/1 April 2022), in collaboration with the Community Nursing Laboratory (Co.Nu.Lab) of the Department of Nursing, University of Thessaly, Greece. The licensing agreement to collect data was obtained by the local municipality authorities (“Help at Home” program of Evrytania—Pr. No. 178/29 April 2022). Before its implementation, the participants provided their written and signed informed consent, fully aware that their participation was voluntary. Also, the process was anonymous, and participants could withdraw from the study at any time in full compliance with the General Data Protection Regulation (GDPR) [EU 2016/679] regarding sensitive personal data.

### 2.4. Statistical Analysis

Descriptive statistics were generated as appropriate for each variable. Categorical variables were summarized as frequencies (n) and percentages (%), while continuous variables were presented as the mean and standard deviation (SD). Shapiro–Wilk’s test was used to assess the normality of quantitative variables. Associations between sociodemographic and health-related characteristics and loneliness symptoms were explored using chi-square tests. A comparison was made between high loneliness (UCLA-3 score > 43) and low/moderate loneliness (UCLA-3 score ≤ 43). Associations between the frailty (TFI) domain score and the loneliness (UCLA-3) score were explored with Pearson’s correlation coefficients. In addition, to investigate the impact of frailty on loneliness, an analysis of covariance (ANCOVA) was performed with the UCLA-3 score as the dependent variable, frailty domains as independent variables, and gender, age, marital status, family status, living status, educational level, annual income, chronic diseases, homebound status, and social isolation level used as covariates. Marginal means, along with 95% confidence intervals (CI), were reported from the analysis of covariance. A *p*-value < 0.05 was preset as statistically significant. Data were encoded and analyzed using the IBM SPSS 26.0 software.

## 3. Results

In Table 1, the demographic characteristics of the study participants are presented. The mean age of the 218 participants (female 61.9%) was 81.45 ± 9, with 60.1% of them aged ≥80 years. In brief, 62% had at least a primary educational level, 53.7% were married, and 85.3% had at least one child. Also, 28.9% lived alone, and 59.6% lived under the poverty threshold. Additionally, 20.6% of the older adults were free from any chronic diseases, while 33% had two chronic diseases, 28.4% had at least three chronic diseases, and 21.1% were homebound.

Frailty was identified in 58.3% of the study participants (TFI mean ± SD: 5.95 ranging 0–13) and were categorized according to TFI domains as follows: physical frailty 57.3%, psychological frailty 43.6%, and social frailty 27.1%. Higher levels of loneliness were identified in 54.1%, with mean UCLA-3 scores of 45.76 ± 11 (range 20–80). Similarly, higher levels of social isolation were identified in 46.3% of our study sample.

Table 2 shows the associations between sociodemographic characteristics and stronger feelings of loneliness. Loneliness was significantly more frequent in females (60% vs. 44.6%, *p* = 0.038), in those aged ≥80 (74.8% vs. 23%, *p* < 0.001), in unmarried participants (37.6% vs. 73.3%, *p* < 0.001), in illiterate older adults (74.7% vs. 28%, *p* < 0.001), in participants with a lower income (61.5% vs. 43.2%, *p* < 0.001), in older adults with multimorbidity (79% vs. 28.9%, *p* < 0.001), in homebound participants (97.8% vs. 27.2%, *p* < 0.001), in frail participants (82.7% vs. 14.3%, *p* < 0.001), and in socially isolated (76.2% vs. 35%, *p* < 0.001) participants.

Table 3 shows the results of the bivariate analysis and Pearson’s correlation coefficient. Particularly, participants with frailty experienced stronger feelings of loneliness across all TFI domains.

In Table 4, we present the effect of frailty on loneliness, using TFI domains as independent variables and socioeconomic factors as covariates and adjusting for potential confounding effects. A higher feeling of loneliness (UCLA-3 mean values) was observed in older adults with physical frailty (48.99 vs. 41.42, *p* < 0.001), in those with psychological frailty (48.6 vs. 43.57, *p* < 0.001), and those with social frailty (53.38 vs. 42.94, *p* < 0.001), compared to non-frail older adults. This association was also observed in frail older adults (total TFI: 49.27 vs. 40.87, *p* < 0.001).

## 4. Discussion

To our knowledge, this is the first study in Greece investigating the impact of frailty on loneliness in terms of socioeconomic status in community-dwelling older people in their homes. Upon analyzing the collected data, 58.3% of the study’s participants were identified as frail, 54.1% experienced strong feelings of loneliness, and 46.3% were characterized as highly socially isolated. Our data analysis revealed that participants with disadvantaged social status (marital status, family, economic, social isolation, and homebound status) experienced significantly higher levels of loneliness. Most importantly, stronger feelings of loneliness were observed in participants with frailty across all domains (physical, psychological, and social).

The main finding of the present study was that frailty was significantly associated with high loneliness, suggesting that frail older adults (>80 years old) with disadvantaged social status experience stronger feelings of loneliness. In agreement with our findings, recent longitudinal studies have shown that respondents with higher levels of loneliness had a higher frailty index score, as did those with a higher level of social isolation. However, the associations remained significant when both loneliness and social isolation were included in a joint model [36]. Along the same line, recent longitudinal associations of loneliness and frailty with up to 20 years of follow-up showed that frailty increased over time and that baseline loneliness predicted frailty progression, highlighting loneliness as a separate construct within social relationships [17]. In addition, results from the previous Longitudinal Aging Study Amsterdam showed that frailty is associated with poor social functioning and with an increase in loneliness over time in community-dwelling older adults [40]. In our study, we recognized the impact of frailty on loneliness in an effect model after adjusting for social disadvantages (social, economic, and homebound statuses). Possible explanations for this association could be that there are interdependencies between loneliness and frailty. For instance, the effect of prior frailty on subsequent loneliness may be greater than the effect of prior loneliness on subsequent frailty in older adults over time, and/or maybe there have been changes in loneliness and changes in frailty over time, according to the China Health and Retirement Longitudinal Study (4 years follow-up) [15]. This may also highlight the possibility of a two-way and mutually reinforcing connection between these two commonly concurrent conditions in old age [41].

We also found strong associations between sociodemographic characteristics and stronger feelings of loneliness, such as a growing intensity of loneliness with advancing age. The high frequency of loneliness in aging may exist due to stereotypes [42]. Additionally, loneliness is largely frequent in older age (>80 years old) [43], which is in agreement with the findings of our study, as 60.1% of the participants were 80 years or older. The higher levels of loneliness in those with a low educational level and poor family support were demonstrated in our study and were consistent with other international studies [2,42,44,45]. Finally, our finding that the high degree of social isolation may be considered a contributing factor to intense loneliness in older age was also identified in other researches [45,46].

Furthermore, the number of chronic diseases that older people suffer from has been found to lead to a stronger feeling of loneliness, which links poor health status and the impact it has on the daily life and functioning of older people with their emotional/psychological state [42]. The connection between the state of health and the level of loneliness is also established by the resulting high rate of loneliness in homebound older adults. Likewise, similar studies revealed the impact of reduced functionality that enhances social networks on older adults’ emotional needs, as can be seen from the degree of loneliness they express [47,48].

Finally, loneliness proves to be an important but often neglected social determinant of quality of life for older people. The World Health Organization addresses loneliness as a pressing public health and policy concern, while the UN Decade of Healthy Aging (2021–2030) has promoted actions to address it. Given the recognized high frequency of psychological and social frailty in older age, as well as their association with adverse health and functional conditions, these two domains of frailty should be seriously taken into account and evaluated in the older population [49,50].

### 4.1. Future Implications

The findings of the present study highlight the importance of interventions tailored to strengthening physical functionality and social engagement and will contribute to combating loneliness among older adults [51,52]. Additionally, the discovery that frailty encompasses not only the physical state of an individual but also incorporates psychological and social elements highlights the necessity for a comprehensive evaluation and multidimensional care approach for the elderly. Remarkably, nurses can identify many causes and consequences of frailty, focusing on the social and psychological domains, which point towards a holistic approach to both loneliness and frailty [53]. Hence, supporting older individuals’ social network and enhancing the education of healthcare professionals, especially community nurses [54,55], are crucial.

### 4.2. Limitations

A potential limitation of the present study is our cross-sectional design, as it does not account for changes in variables over time. Nevertheless, we recognize that longitudinal research may be undertaken for the testing of changes in variables over time. We also cannot determine causality due to the observational nature and single-point data collection, for instance, to explain what caused those correlations. Although TFI and UCLA-3 have been validated for the Greek population, the responses of the participants consist of self-witnesses, which may not be accurate, due to lack of self-awareness. Another potential limitation could be the low response rate that may bias our results. To counteract the bias, we applied the analysis of covariance and controlled for interaction effects (covariates), meaning that any bias from a low response rate was eliminated [56]. Also, this study was conducted during the COVID-19 pandemic, and therefore, participants’ responses might have been impacted by the lockdown consequences. Nevertheless, examining these factors during this particular time frame yielded valuable insights into understanding the physical and psychosocial well-beings of seniors in challenging pandemic conditions.

## 5. Conclusions

Our study findings indicate that frail older adults experience stronger feelings of loneliness across all frailty domains (physical, psychological, and social), independently of their socioeconomic status and existing comorbidities. Thus, more attention should be given to early intervention for frailty and the continuous screening of both frailty and social vulnerabilities in older adults. Given the relevance of social frailty (domain) to social disadvantages, including social isolation in terms of economic situation, family support, and psychosocial factors of older adults, further research is warranted to explore the mechanisms and causal pathways of the association between loneliness and frailty. Thus, it is important for health professionals caring for older people to pay more attention to social disadvantages in designing effective interventions to enhance the well-being of older adults.

## Figures and Tables

**Table 1 healthcare-12-01666-t001:** Descriptive characteristics of the participants (n = 218).

Characteristics	Categories	n	%
Sex	Male	83	38.1
	Female	135	61.9
Age (years) [mean ± SD: 81.45 ± 9.06]	65–79	87	39.9
	≥80	131	60.1
Marital Status	Married	117	53.7
	Single/Divorced/Widowed	101	46.3
Family Status (Having Children)	Yes	186	85.3
	No	32	14.7
Living Alone	Yes	63	28.9
	No	155	71.1
Educational Level	Illiterate	83	38.1
	Primary	110	50.5
	Secondary/Tertiary	25	11.5
Annual Individual Income (Euros)	<5.269	130	59.6
	>5.269	88	40.4
Chronic Diagnosed Diseases (number)	0	45	20.6
	1	39	17.9
	2	72	33.0
	≥3	62	28.4
Homebound Status	Non-Homebound	114	52.3
	Semi-Homebound	58	26.6
	Homebound	46	21.1
Frailty (TFI) [mean ± SD: 5.95 ± 3.07]	Physical Frailty (Yes)	125	57.3
	Psychological Frailty (Yes)	95	43.6
	Social Frailty (Yes)	59	27.1
	Total Frailty (Yes)	127	58.3
Social Isolation	Low	28	12.8
	Moderate	89	40.8
	High	101	46.3
Loneliness (UCLA-3) [mean ± SD: 45.76 ± 11.10]	Low	10	4.6
	Moderate	90	41.3
	High	118	54.1

Abbreviations: SD: standard deviation; TFI: Tilburg Frailty Indicator; UCLA-3: UCLA Loneliness Scale (Version-3). Notes: Data were presented as actual numbers (n) and percentages (%); annual individual income refers to the poverty threshold cut-off criterion; chronic diagnosed diseases contain the most common chronic diseases according to participants’ medical records. Homebound refers to the ability to leave the home in the last month.

**Table 2 healthcare-12-01666-t002:** Associations between sociodemographic characteristics and higher feelings of loneliness (n = 118).

Characteristics	Categories	Older Adults with Higher Feelings of Loneliness/Total (%)	Test χ^2^	*p*-Value
Sex	Male	37/83 (44.6)	4.322	0.038
	Female	81/135 (60.0)		
Age (years)	65–79	20/87 (23.0)	54.474	<0.001
	≥80	98/131 (74.8)		
Marital Status	Married	44/117 (37.6)	26.345	<0.001
	Unmarried	74/101 (73.3)		
Having Children	Yes	100/186 (53.8)	0.005	0.945
	No	18/32 (56.3)		
Living Alone	Yes	45/63 (71.4)	9.723	0.002
	No	73/155 (47.1)		
Educational Level	Illiterate	62/83 (74.7)	25.087	<0.001
	Primary	49/110 (44.5)		
	Secondary/Tertiary	7/25 (28.0)		
Annual Individual Income	<5.269 Euros	80/130 (61.5)	6.402	0.011
	>5.269 Euros	38/88 (43.2)		
Number of Chronic Diseases	0	13/45 (28.9)	27.502	<0.001
	1	17/39 (43.6)		
	2	39/72 (54.2)		
	≥3	49/62 (79.0)		
Homebound Status	Non-Homebound	31/114 (27.2)	76.497	<0.001
	Semi-Homebound	42/58 (72.4)		
	Homebound	45/46 (97.8)		
Physical Frailty	Yes	100/125 (80.0)	76.564	<0.001
	No	18/93 (19.4)		
Psychological Frailty	Yes	79/95 (83.2)	55.092	<0.001
	No	39/123 (31.7)		
Social Frailty	Yes	46/59 (78.0)	17.220	<0.001
	No	72/159 (45.3)		
Total Frailty	Yes	105/127 (82.7)	97.132	<0.001
	No	13/91 (14.3)		
Social Isolation	Low/Moderate	41/117 (35.0)	35.408	<0.001
	High	77/101 (76.2)		

Notes: Data were given as actual numbers of older adults and percentages (%); higher feelings of loneliness were identified in those with a UCLA-3 mean value ≥ 43.

**Table 3 healthcare-12-01666-t003:** Associations between frailty (TFI) and loneliness (UCLA-3).

TFI Domains Score	UCLA-3 Loneliness Scale Score
Physical Frailty	0.719 *
Psychological Frailty	0.651 *
Social Frailty	0.547 *
Total Frailty	0.897 *

Notes: methods: bivariate analysis and Pearson’s correlation coefficient. * Significant level represented by *p* < 0.001.

**Table 4 healthcare-12-01666-t004:** Effect of frailty on high loneliness.

TFI Domains	Marginal Means (95% C.I.)	*F*	*p*-Value	*η* ^2^
Physical Frailty		23.672	<0.001	0.105
No	41.42 (39.48–43.37)			
Yes	48.99 (47.44–50.54)			
Psychological Frailty		25.117	<0.001	0.111
No	43.57 (42.37–44.77)			
Yes	48.60 (47.20–49.99)			
Social Frailty		27.431	<0.001	0.120
No	42.94 (41.59–44.29)			
Yes	53.38 (50.39–56.36)			
Total Frailty		28.594	<0.001	0.125
No	40.87 (38.89–42.86)			
Yes	49.27 (47.73–50.80)			

Abbreviations: C.I: confidence intervals. Notes: Method: analysis of covariance (ANCOVA); UCLA-3 score was used as the dependent variable; frailty domains were used as independent variables; gender, age, marital status, family status, living status, educational level, annual income, chronic diseases, homebound status, and social isolation level were used as covariates.

## Data Availability

The data presented in this research are available upon request from the corresponding author. The data are unavailable to the public due to privacy limitations.

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
