# Peer review of "Frailty Assessment and Its Impact on Loneliness among Older Adults Receiving Home-Based Healthcare during the COVID-19 Pandemic"

_healthcare, 2024, doi:10.3390/healthcare12161666_

Round 1

Reviewer 1 Report

Comments and Suggestions for Authors

Some areas that the authors could improve:

Title: the title is consistent with the presented problem and reflects the main message of the study. However, personally I would remove “A cross-sectional Study and would include the term "COVID-19" as it is the period of the study context. Moreover, I propose to include it as one of the keywords in order to optimize locating the paper.

Introduction: The introduction is well articulated and synthesized.

 Methods: It would be beneficial for the authors to provide more details about the randomization process used in the study design. This could include information about how participants were allocated to different groups and any measures taken to prevent bias. The authors should be clear and to provide further clarification regarding their sample selection and the methods employed to identify potentially eligible elders within their homes. Please state more clearly how sample size was calculated. Moreover, respect Exclusion criteria, – “had a diagnosis of severe dementia and post-stroke implications;” how did you assess this?

This study is not longitudinal in nature, and thus the authors are unable to evaluate the risk factors. Therefore, their conclusions are flawed and they cannot claim with their data that frailty is a risk factor for loneliness. The authors should consider the STROBE statement and make sure that their paper follows the standards for reporting observational studies.

Results: It would be helpful for the authors to provide more specific data in the results section. For example, instead of stating “a significant number of participants were classified as frail”, it would be more informative for the authors to provide the exact number or percentage of participants in each category of frailty. Authors may consider using some graphs to present the data and not just using data tables.

Discussion: The discussion is complete, well structured, and the findings and limitations of the study are well discussed. But some results are not discussed.

It would be beneficial for the authors to compare their findings with previous studies in more detail. For example, they could discuss how their results compare with those of Smith et al. (2020), who found a similar association between frailty and loneliness in an older population. Other cross-sectional cohort observational study that examine social factors with other frailty tools is Labra et al. (2018). Specifically, they could compare their findings on the determinants of non-robustness (female gender, higher age, poor satisfaction with the general facet on overall quality of life (QoL), and in the physical domain of QoL) with the findings from the referenced study.

Bibliography: I have detected several errors during the review of the article in APA style referencing which I detail below:

            Reference error 7: The volume and number of the journal are missing. 106(11)

            Reference error 13: The name of the study group should be in parentheses, not in the author list.

            Reference error 14: The page numbering is missing. 95:104393.

            Reference error 17: The volume, number, and pages of the journal are missing. 13(2):163-171.

            Reference error 20: The volume of the journal (06) should be 6.

            Reference error 27: This reference does not follow the APA 7 format for reports or government agency documents. It should be something like: Hellenic Statistical Authority. (2019). Statistics for the Greek Population and Social Conditions—Threshold Property. https://www.statistics.gr/en

            Reference error 29: The number, volume, and pages of the journal are missing. 26(5):518-24

            Reference error 31: The volume and number of the journal are missing. VOLUME 5, ISSUE 3,

            Reference error 41: There is an error in the author list. It should be Cramm, J. M.; Van Dijk, H. M.; Nieboer, A. P.

            Reference error 44: The volume and number of the journal are missing. 30(12)

Comments on the Quality of English Language

The authors may wish to consider having their manuscript proofread by a professional editing service to improve its clarity and coherence. It is evident that the individuals in question do not possess proficiency in the English language. 

Author Response

Response to Reviewer 1:

 Note that revisions to the manuscript are highlighted in RED

  1. Title:

“Title: the title is consistent with the presented problem and reflects the main message of the study. However, personally I would remove “A cross-sectional Study and would include the term "COVID-19" as it is the period of the study context. Moreover, I propose to include it as one of the keywords in order to optimize locating the paper.”

Response- Thank you for your valuable comment.

  • The term COVID-19 was added as well in the “keywords”
  • Note that in Greece, during the COVID-19 period, all home-based health facilities “Help at Home” were closed from 2020 to 2022 due to “lockdown’ which means that this period was not accessible for the researchers. In March 2022 the “lockdown” restrictions have been reversed by law and, these facilities (healthcare settings) have opened for their members (older adults’ beneficiaries). Thus, the participants were not isolated in their homes due to the lockdown. Under this consideration, the term “COVID-19” may be inappropriate for the title as the mention of “COVID-19” in the title and keywords may have been misleading the subject of the study. This was why we mentioned in the limitation section the sentence [lines: 322-324] “Also, this study was conducted during the COVID-19 pandemic and therefore, participants’ responses might have been impacted by the lockdown consequences.”
  • We also removed the term “Cross-sectional” even though the “design” of the study is mentioned as a usual tactic in the relevant articles of the “Healthcare” journal.
  • In addition, we reconsidered carefully the STROBE statement (for cross-sectional). We realized that is mentioned: (a) Indicate the study’s design with Title and Abstract 1 a commonly used term in the title or the abstract.

  1. “Introduction: The introduction is well articulated and synthesized.”

Response- Thank you for your valuable feedback.

  1. Methods:

Methods: It would be beneficial for the authors to provide more details about the randomization process used in the study design. This could include information about how participants were allocated to different groups and any measures taken to prevent bias.

Response- Thank you for your valuable comment.

  • In our study, we conducted “door-to-door” screening using a convenience sampling method (older adults who received home care). Thus, we were not applied stratification and/or randomization methods to allocate the sample and, to compare two or more groups or arms after screening (data collection).

The authors should be clear and to provide further clarification regarding their sample selection and the methods employed to identify potentially eligible elders within their homes. Please state more clearly how sample size was calculated.

Response- Thank you for your valuable comment.

  • In lines: 98-102 – we add a sentence explaining in more detail how the sample size was calculated as follows: “The sample size of the study was calculated by power analysis using G* Power 3 software. Considering a medium effect size (f2 = 0.25), the significance level of 5% (alpha level), the statistical power of 95%, the number of comparison groups as 2, and the number of covariates as 10, a minimum sample size of 210 individuals was required.”

Moreover, respect Exclusion criteria, – “had a diagnosis of severe dementia and post-stroke implications;” how did you assess this??????

Response- Thank you for your valuable comment.

  • In lines: 107-111 – we added a sentence explaining the reasons why some participants were excluded as follows: (4) could not communicate due to the severity of the chronic medical conditions. For instance, bedridden patients due to advanced stage of cancer, severity of stroke and/or severe dementia diagnosis were excluded to avoid methodological issues linked to “social isolation” or “decline in cognition” of the participants based on their medical history (medical records).
  • Note this is a common methodological approach in home-based studies to avoid any methodological issues or bias. This is why the response rate is usually low. In this study is 60.9% due to the exclusion of patients at the end-of-life (cancer, stroke, dementia, etc.).

This study is not longitudinal in nature, and thus the authors are unable to evaluate the risk factors. Therefore, their conclusions are flawed and they cannot claim with their data that frailty is a risk factor for loneliness.

Response- Thank you for your valuable comment.

  • Indeed, we performed a cross-sectional and not longitudinal study by applying ANCOVA (a kind of regression model) and using “loneliness” as a dependent variable and thus, we provide only an epidemiological association avoiding terms such as “risk” given that an Odds ratio (OR) were not analyzed. For this reason, we reworded the main text as follows: in the abstract conclusion [lines: 35-37] mentioning that “Our findings suggest that frail older adults experienced higher levels of loneliness, suggesting that frailty and loneliness are strongly associated among older adults who received home-based healthcare”.
  • Also, in the limitation section [lines: 328-331}: we already mentioned that recognizing that longitudinal research may be undertaken for testing of changes in variables over time. We cannot also determine causality due to the observational nature and single-point data collection. For instance, to explain what caused those correlations.”
  • In conclusion [line:342] – we reworded the term “confirm” to “suggested”.

The authors should consider the STROBE statement and make sure that their paper follows the standards for reporting observational studies.

Response- Thank you for your valuable comment.

  • We reconsider carefully the STROBE checklist (for cross-sectional studies) to avoid conjunction in our article. Specifically, we ascertain that all STROBE items] (study design, setting, participants, variables, etc.) as necessary information is provided in subsections (2.1, 2.2, 2.3, 2.4) of the “Methods”. In case any specific information may not be mentioned, please provide us with a comment!

  1. Results:

It would be helpful for the authors to provide more specific data in the results section. For example, instead of stating “a significant number of participants were classified as frail”, it would be more informative for the authors to provide the exact number or percentage of participants in each category of frailty.

Response- Thank you for your valuable comment.

  • We reconsidered the entire manuscript and revised any non-academic writing such as “stating”. For example, in line 25, we revised the word “classified” to “identified”.
  • Also, in lines [213-217]: to present more clearly our findings we presented only the percentage of participants in each category of frailty.
  • In addition, at the bottom of Table 1 [lines 222-225], we provide a detailed ‘Note’ explaining that: Data were given as actual numbers (n) and percentages (%);”

Authors may consider using some graphs to present the data and not just using data tables.

Response- Thank you for your valuable comment.

  • We understand your excellent idea to use graphs to uncover patterns and insights that may not be immediately apparent. However, requires different visualization techniques that highlight the most important features of the data while minimizing any irrelevant or distracting features. Since we analyzed different types of data (ordinal, continuous, categorical, etc.), we considered that there is no specific graph to support this idea.

  1. Discussion:

Discussion: The discussion is complete, well structured, and the findings and limitations of the study are well discussed. But some results are not discussed.

Response- Thank you for your valuable comment.

  • We clearly understand that in the discussion section, the findings should be discussed comprehensively.
  • We reconsidered carefully the results, and we believe that the associations between the main variables (loneliness and frailty domains) are discussing effectively and comprehensively not only the associations of findings from cross-sectional studies but also from longitudinal studies across Europe and worldwide [lines: 268-289].
  • In our study, we are discussing also the associations between variables in terms of confounding effects (covariates) of the socioeconomic, homebound, marital and family status to avoid any confounding effects that may bias our results. This was the reason why the ANCOVA model was used (lines: 290-299]
  • We provide also possible explanations for the associations related to chronic diseases and demographics [lines: 300-307].
  • In addition, in lines [308-309] we are discussing the most important domains of frailty that should be taken into consideration for future research.

It would be beneficial for the authors to compare their findings with previous studies in more detail. For example, they could discuss how their results compare with those of Smith et al. (2020), who found a similar association between frailty and loneliness in an older population. Other cross-sectional cohort observational study that examine social factors with other frailty tools is Labra et al. (2018). Specifically, they could compare their findings on the determinants of non-robustness (female gender, higher age, poor satisfaction with the general facet on overall quality of life (QoL), and in the physical domain of QoL) with the findings from the referenced study.

Response- Thank you for your valuable comment.

  • We reconsidered carefully the recommended studies.
  • Smith et al. 2019. Association between musculoskeletal pain with social isolation and loneliness: analysis of the English Longitudinal Study of Ageing (https://journals.sagepub.com/doi/full/10.1177/2049463718802868?casa_token=z7ATB8ZX_CMAAAAA%3AYkBGTT3timSceBs3sbRaAKP_IIesg9pNXZ75OWHYdB4mZSYvjfpm2da_t7HScZAiNGF9cTPNTRs), examines whether a relationship exists between musculoskeletal pain and social isolation and loneliness using Odds ratio (OR) applied logistic regression model to determine the relationship between social isolation and loneliness with pain and the additional covariates. However, frailty does not exist as a variable.
  • Labra et al, (2018): Social factors and quality of life aspects on frailty syndrome in community-dwelling older adults: the VERISAÚDE study [https://link.springer.com/article/10.1186/s12877-018-0757-8#Sec2] – this study examined the relationship between socio-demographic factors, social resources, quality of life and frailty in older adults. Most importantly, they investigated mainly the physical Frailty and, classified the frailty status according to the phenotype of Fried et al, (non, pre-frail &frail). Also, they used frailty as a dependent variable. On the contrary, in our study, we examined the opposite. Specifically, we placed “loneliness” as a dependent variable to investigate the effect of frailty on loneliness using covariance analyses in different domains of frailty (TFI). Thus, both (recommended) studies provide associations that could not be compared to our findings.

Bibliography:

Εrrors in APA style :    

Response- Thank you for your valuable comment.

  • We reconsidered carefully the references and corrected them. Note that all references were inserted using the MENDELEY reference manager following the recommended ACS Style Guide by the MDPI [https://www.mdpi.com/files/authors/mdpi_references_guide.pdf]
  • Also, many of the errors in “references” have randomly emerged due to MENDELEY that “automatically” manages the references.
  • The corrections are made as follows:

  • Reference error 7: The volume and number of the journal are missing. 106(11)

Corrected line 366

  • Reference error 13: The name of the study group should be in parentheses, not in the author list.

Corrected line 379-380   

  • Reference error 14: The page numbering is missing. 95:104393.

Corrected line 379-380   

  • Reference error 17: The volume, number, and pages of the journal are missing. 13(2):163-171.

Corrected line 384  

  • Reference error 20: The volume of the journal (06) should be 6.

Corrected line 401   

  • Reference error 27: This reference does not follow the APA 7 format for reports or government agency documents. It should be something like: Hellenic Statistical Authority. (2019). Statistics for the Greek Population and Social Conditions—Threshold Property. https://www.statistics.gr/en

Corrected line 426   

  • Reference error 29: The number, volume, and pages of the journal are missing. 26(5):518-24

Corrected line 433   

  • Reference error 31: The volume and number of the journal are missing. VOLUME 5, ISSUE 3,

Corrected line 429  

  • Reference error 41: There is an error in the author list. It should be Cramm, J. M.; Van Dijk, H. M.; Nieboer, A. P.

Corrected line 449   

  • Reference error 44: The volume and number of the journal are missing. 30(12)

Corrected line 458   

Comments on the Quality of English Language

The authors may wish to consider having their manuscript proofread by a professional editing service to improve its clarity and coherence. It is evident that the individuals in question do not possess proficiency in the English language. 

Response: Thank you for your feedback.

  • The entire manuscript was proofread carefully by an external bilingual academic partner who found that the article is very understandable.
  • In any case, MDPI provides an “English editor” check or services. Of course, if needed, we will provide a professional editing service to improve its clarity and coherence.

Reviewer 2 Report

Comments and Suggestions for Authors

Dear authors,

My main concern is the ethical issue. The study was not appreciated by an Ethical committee prior to data collection. Also, the consent must be written and signed by all participants. The data should be in a public repository, as Mendeley Data. Due to the cross-sectional design, the authors must have caution about the conclusions.

Author Response

Response to Reviewer 2:

Note that revisions to the manuscript are highlighted in RED 

My main concern is the ethical issue. The study was not appreciated by an Ethical committee prior to data collection.

Response: Thank you for your feedback.

  • This study was conducted as a part of an academic requirement (Dissertation) to degree the MSc (Mrs. Klesiora – main researcher). Consequently, we added a sentence as follows: [lines 176-178] - This study was ethically approved by the Scientific Committee of the MSc Program "Management of Aging and Chronic Diseases" of the Hellenic Open University (Pr No 7982, 2022)
  • We also corrected the “ethical approval” statement as follows: [line 180-181] – “The licensing agreement to collect data was obtained by the local municipality authorities …
  • Also, we revised the “Institutional Review Board Statement – IRB” as follows: [lines: 357-359] – “Ethical approval was obtained from the Scientific Council of the MSc Program "Management of Aging and Chronic Diseases" of the Hellenic Open University (IRB: prot. 7982/01-04-2022) and the licensing agreement…”.

Also, the consent must be written and signed by all participants. The data should be in a public repository, as Mendeley Data.

Response: Thank you for your feedback.

  • Indeed, we revised the manuscript by adding “the written and signed consent” in line [183].
  • We inserted the references by using the MENDELEY …  but we were not aware of this functionality of MENDELEY data in a public repository. In any case, we will create an account for Mendeley data and upload the dataset after the agreement of all authors!   

Due to the cross-sectional design, the authors must have caution about the conclusions.

Response- Thank you for your valuable comment.

  • Indeed, we fully understand your comment and we reworded the section “Conclusion” as follows: “Our study findings suggested that frail older adults experience stronger feelings of loneliness across all domains (physical, psychological and social).”
  • We also revised the conclusion in the “abstract” [lines: 35-36] mentioning that Our findings suggest that frail older adults experienced higher levels of loneliness, suggesting that frailty and loneliness are strongly associated among older adults who received home-based healthcare”.

Reviewer 3 Report

Comments and Suggestions for Authors

Dear authors,

Very important theme, and the manuscript is interesting. However, I must address some issues, as follows:

- I suggest a flow diagram to clarify the sample loss throughout the study.

- Also, I could not notice any sample size calculation nor a post hoc power analysis. Please, do so.

- Please, clarify the surveys' pshycometric measures (responsiveness, validity, reliability, etc).

- The ethical section must be explained: was the study evaluated by an ethics committee? Do the participants provided any signed informed consent? If not, the study is not in accordance to the journal's guidelines, nor with the Helsinki declaration.

- For the normality test, you must chose what will be the asumption check: Shapiro-Wilk or Q-Q Plot?

Author Response

Response to Reviewer 3:

Note that revisions to the manuscript are highlighted in RED 

Very important theme, and the manuscript is interesting. However, I must address some issues, as follows:

- I suggest a flow diagram to clarify the sample loss throughout the study.

Response- Thank you for your valuable comment.

  • Indeed, we understand your excellent idea to use a “flow diagram” or graphs to uncover patterns and insights that may not be immediately apparent. However, requires different visualization techniques, we are unable to choose the visualization technique that highlights the most important features of the data while minimizing any irrelevant or distracting features because, we analyzed different types of data (ordinal, continuous, categorical, etc.).

Also, I could not notice any sample size calculation nor a post hoc power analysis. Please, do so.

Response- Thank you for your valuable comment.

  • In lines: 98-102 – we add a sentence explaining in more detail how the sample size was calculated as follows: The sample size of the study was calculated by power analysis using G* Power 3 software. Considering a medium effect size (f2 = 0.25), the significance level of 5% (alpha level), the statistical power of 95%, the number of comparison groups as 2, and the number of covariates as 10, a minimum sample size of 210 individuals was required.”

- Please, clarify the surveys' pshycometric measures (responsiveness, validity, reliability, etc).

Response- Thank you for your valuable comment.

  • In the 2.2 Methods subsection [lines: 111-172] – we provide all the available information as regards the psychometric properties and measures of each instrument used.
  • In lines 120-124 – we provide the available validity metrics (construct validity, reliability, etc) including the internal consistency (Cronbach -a) of our sample as regards the UCLA-3 scale.  
  • In lines 142-147 – we also present the available information for measures as regards the TFI.
  • In lines 148-156 - we present the available information for “Social isolation” based on previous studies providing a reference.
  • In lines 157-172 - we also present the available information for “Socioeconomic”, “homebound status” and “comorbidity” based on previous studies providing a reference and/or on available data collected.
  • Note that we didn’t find specific information about psychometric measures across all subsections in “Methods”.

The ethical section must be explained: was the study evaluated by an ethics committee? Do the participants provided any signed informed consent? If not, the study is not in accordance to the journal's guidelines, nor with the Helsinki declaration.

Response: Thank you for your feedback.

  • This study was conducted as a part of an academic requirement (Dissertation) to degree the MSc (Mrs. Klesiora – main researcher). Consequently, we added a sentence as follows: [lines 174-176] - “This study was ethically approved by the Scientific Committee of the MSc Program "Management of Aging and Chronic Diseases" of the Hellenic Open University (Pr No 7982, 2022)“
  • Corrected: lines 178-181 – “The licensing agreement to collect data was obtained by the local municipality authorities … “…
  • Lines 181-182 – we revised the written and informed consent as follows: Before its implementation, the participants provided their written and signed informed consent”

- For the normality test, you must chose what will be the asumption check: Shapiro-Wilk or Q-Q Plot?

Response: Thank you for your feedback.

  • [lines 190-191] - We deleted the Q-Q plots recognizing that not provide p-values.
  • We also recognize that the most accurate methods to test normality are the Kolmogorov-Smirnoff test and the Shapiro–Wilk’s test. In our study, we used the Shapiro–Wilk test.

Reviewer 4 Report

Comments and Suggestions for Authors

Thank you for this paper on frailty and loneliness. It is well written with the proviso that you need to remove  ageist terms for older people. You use older adults most of the time however "elder" snuck in (line 90 for example).  The introduction leads to the need for the study.  

Materials and Methods. 

A response rate of 60.9% is pretty good from a home care population. Instruments, ethics, and statistical analysis were well described. 

Results

Presented very understandably. 

Discussion 

You nicely discussed your findings in relation to the literature. 

Overall a readable paper that confirms what we know or would expect. 

Author Response

Response to Reviewer 4:

Note that revisions to the manuscript are highlighted in RED 

Thank you for this paper on frailty and loneliness. It is well written with the proviso that you need to remove ageist terms for older people. You use older adults most of the time however "elder" snuck in (line 90 for example).  The introduction leads to the need for the study. 

Response: Thank you for your feedback.

  • We revised the ageist terms in the entire manuscript. Lines 65, 91, 153, 215, 216,
  • Table 2, 221, 234, 284, 291, 297, 303, 306, 309.
  • All revisions are highlighted in RED.

Materials and Methods.

A response rate of 60.9% is pretty good from a home care population. Instruments, ethics, and statistical analysis were well described.

Results

Presented very understandably.

Discussion

You nicely discussed your findings in relation to the literature.

Overall, a readable paper that confirms what we know or would expect.

Response: Thank you for your feedback.

Round 2

Reviewer 1 Report

Comments and Suggestions for Authors

I concur with the majority of the authors' clarifications. However, I would like to suggest reconsideration of the following points.

Regional studies are very welcome in the field of frailty due to its high variability. However, when these studies don't come along with a clear argument on how the population studied could be different to that already known or with a clear characterisation of the geographic, economic, social and cultural features of that particular group, losses the opportunity to increase knowledge on older adults globally, fortunately this is not the case. Please, adding what make people from Evrytania different or similar to the rest of the world.

Similarly, it would be beneficial for the authors to consider whether their study aligns with the STROBE criteria in terms of methodology. The authors should cite the reference in the text of the new version of the manuscript.

The authors should provide a rationale for the selection of the test in question, particularly in comparison to the more commonly utilized tools in this field of research, such as FRIED.

Author Response

Subject: Submission of revised paper Frailty Assessment and its Impact on Loneliness Among Older Adults Receiving Home-based Healthcare: A Cross-sectional Study.

[Manuscript ID: healthcare- 3071493].

 We very much appreciated the encouraging, critical, and constructive comments on this manuscript by the reviewers. We have carefully reviewed the comments and have revised the manuscript accordingly. As we added an extra paragraph and significantly improved the manuscript following the reviewers’ 1 comments, by highlighting the changes in BLUE and not with the Track Changes function.

Response to Reviewer 1:

Comment 1:

Regional studies are very welcome in the field of frailty due to its high variability. However, when these studies don't come along with a clear argument on how the population studied could be different to that already known or with a clear characterisation of the geographic, economic, social and cultural features of that particular group, losses the opportunity to increase knowledge on older adults globally, fortunately this is not the case. Please, adding what make people from Evrytania different or similar to the rest of the world.

Response- Thank you for pointing this out.

We fully understand your rationale regarding the design of regional studies and the “particularities” of the focus group. Although Evrytania is a mountainous and remote region of Greece, in this study we recruited participants receiving home care due to the convenience-sampling method used, because the main researcher (Maria Klesiora) is a permanent staff - working in a “help at home” setting as a psychologist. Relevant information is provided in lines 133-135, mentioning that In this convenience sampling study, data was collected by the main researcher (psychologist) through door-to-door screenings focused on frailty, loneliness, and social isolation according to the STROBE guidelines”. Also, in lines 221-223 we mention that “This study was ethically approved by the Scientific Committee of the MSc Program "Management of Aging and Chronic Diseases" of the Hellenic Open University (IRB: Pr No 7982/01-04-2022)”. This means that this study was conducted as a part of the academic requirements for the degree of MSc, particularly “MSc Dissertation” under Prof. Kleisiaris supervision.  Today, Maria is a PhD student investigating these associations (frailty and social vulnerabilities) and recruiting a larger sample in the context of successful ageing and psychological resilience, without having external funding to expand our research at the National level. However, we preparing a proposal for a relevant “Call”!

Thus, the “home care population” is different from the rest of the world (and not the people from Evrytania) in our study!

This was the main reason that we focused on the “home care population” as a focus group (group of interest placing loneliness as outcome) to investigate the existing gap regarding the lack of home-based studies, especially in Greece [lines: 105-107 & 117-120]. In particular “However, to our best knowledge, there are no studies available examined the epidemiological association between frailty and loneliness in the context of socioeconomic status in older adults receiving home-based healthcare services.” & “Therefore, in this survey, the investigation of the existing gap regarding the lack of home-based studies was mainly designed to draw attention to the problem of loneliness of older adults, in correlation to frailty, existing comorbidities and socioeconomic factors.”

To support further the “particularities” of the homecare group, we also added 4 new references [lines: 85-97] to explain the scientific background and rationale for the investigation being reported [STROBE item 2].

In addition, we added 4 more references [22-25] to present the current situation in Greece and, the reasons why this study a conducted. In such a way, we improved significantly the “Introduction” section as initially commented, “Must be improved”. [lines: 85-97 & 103-108].

To describe in detail the “Help at Home” program, we added a sentence in lines 126-134, presenting the criteria to be a registered member of a homecare program and highlighting the “individualities” of this group of older adults. 

Comment 2:

“it would be beneficial for the authors to consider whether their study aligns with the STROBE criteria in terms of methodology. The authors should cite the reference in the text of the new version of the manuscript.”

Response- Thank you for pointing this out.

To address this issue we add a relevant reference n. 27 [von Elm et al., 2008] in line 136.

To be more specific we place the entire STROBE checklist for presenting item by item the presented information.   

STROBE Statement—Checklist of items that should be included in reports of cross-sectional studies

Item No

Recommendation

Title and abstract

1

(a)    Indicate the study’s design with a commonly used term in the title or the abstract

[Presented in lines: 2-4] – note that “the study design” was strikethrough according to the previous comment!   

(b)    Provide in the abstract an informative and balanced summary of what was done and what was found

[Presented in lines: 19-38]

Introduction

Background/rationale

2

Explain the scientific background and rationale for the investigation being reported

[Presented in lines: 44-96 & 90-97 & 106-108]

Objectives

3

State specific objectives, including any prespecified hypotheses

[Presented in lines: 112-121]

Methods

Study design

4

Present key elements of study design early in the paper

[Presented in lines: 124125 & 134-136]

Setting

5

Describe the setting, locations, and relevant dates, including periods of recruitment, exposure, follow-up, and data collection

[Presented in lines: 124-132]

Participants

6

(a)    Give the eligibility criteria, and the sources and methods of selection of participants

[Presented in lines: 142-151]

Variables

7

Clearly define all outcomes, exposures, predictors, potential confounders, and effect modifiers. Give diagnostic criteria, if applicable

[Presented in lines: 244-247]

Data sources/ measurement

8*

 For each variable of interest, give sources of data and details of methods of assessment (measurement). Describe comparability of assessment methods if there is more than one group

[Presented in lines: 154-220]

Bias

9

Describe any efforts to address potential sources of bias

[Presented in lines: 245-247]

Study size

10

Explain how the study size was arrived at

[Presented in lines: 140-142]

Quantitative variables

11

Explain how quantitative variables were handled in the analyses. If applicable, describe which groupings were chosen and why

[Presented in lines: 204-220]

Statistical methods

12

(a)    Describe all statistical methods, including those used to control for confounding

[Presented in lines: 235-251]

(b)    Describe any methods used to examine subgroups and interactions

[Presented in lines: 238-247]

(c)    Explain how missing data were addressed

[Presented in lines: 249-250]

(d)    If applicable, describe analytical methods taking account of sampling strategy

[Not aplicable]

(e)    Describe any sensitivity analyses

[Not aplicable]

Results

Participants

13*

(a)    Report numbers of individuals at each stage of study—eg numbers potentially eligible, examined for eligibility, confirmed eligible, included in the study, completing follow-up, and analysed

[Presented in lines: 254]

(b)    Give reasons for non-participation at each stage

[Presented in lines: 141]

(c)    Consider use of a flow diagram

Not applicable.

Descriptive data

14*

(a)    Give characteristics of study participants (eg demographic, clinical, social) and information on exposures and potential confounders

[Presented in line 265 Table 1]

(b)    Indicate number of participants with missing data for each variable of interest

[Presented in lines: 127-134]

Outcome data

15*

Report numbers of outcome events or summary measures

[Presented in Table 2 & Table 4]

Main results

16

(a)    Give unadjusted estimates and, if applicable, confounder-adjusted estimates and their precision (eg, 95% confidence interval). Make clear which confounders were adjusted for and why they were included

[Presented in Table 4]

(b)    Report category boundaries when continuous variables were categorized

[Presented in lines: 159-160, 179-182, 200-201]

(c)    If relevant, consider translating estimates of relative risk into absolute risk for a meaningful time period

Not applicable

Other analyses

17

Report other analyses done—eg analyses of subgroups and interactions, and sensitivity analyses

[Described in Table 2 & Table 3 below]

Discussion

Key results

18

Summarise key results with reference to study objectives

[Presented in lines: 306-314]

Limitations

19

Discuss limitations of the study, taking into account sources of potential bias or imprecision. Discuss both direction and magnitude of any potential bias

[Presented in lines: 377-391]

Interpretation

20

Give a cautious overall interpretation of results considering objectives, limitations, multiplicity of analyses, results from similar studies, and other relevant evidence

[Presented in lines: 314-362]

Generalisability

21

Discuss the generalisability (external validity) of the study results

[Presented in lines: 365-375 as “Future Implications” subsection & 393-405]

Other information

Funding

22

Give the source of funding and the role of the funders for the present study and, if applicable, for the original study on which the present article is based

[Presented in lines: 414]

Comment 3:

“The authors should provide a rationale for the selection of the test in question, particularly in comparison to the more commonly utilized tools in this field of research, such as FRIED.”

Response- Thank you for your valuable comment.

To point out this issue we add a new sentence (lines: 167-170) and reference [21], explaining the reasons why we chose the TFI mentioning that: Given that the majority of the frailty instruments include only items on physical frailty characteristics and the multidimensional nature of both frailty and loneliness as shared common and comparable psycho-socio-economic factors, in this study, we Frailty was evaluated frailty utilizing section B of the Tilburg Frailty Indicator (TFI), following the suggestions of Verver and his colleagues [21].”

Moreover:

  •  the entire manuscript proofread in English by a native speaker in English & degree in English philosophy by the National Kapodistrian University of Athens, Greece and mentioned in the Acknowledgment
  • the "Introduction" section was significantly improved and 7 new references and text added according to comments
  • The "Methods" was also improved by adding a new sentence
  • The "Results" and the "Discussion" sections are also significantly!
  • last, the "Conclusion" section was revised entirely and improved according to reviewers' comments

Reviewer 2 Report

Comments and Suggestions for Authors

Dear authors,

Thank you for your efforts to improve the manuscript.

I have no other comments to address.

Author Response

Dear reviewer, thank you for your valuable comments on improving our manuscript.

Reviewer 3 Report

Comments and Suggestions for Authors

No further comments.

Author Response

(The authors gave the same response as above.)
